# Latent Association Between Diets and Glioma Risk: A Mendelian Randomization Analysis

**DOI:** 10.3390/nu17030582

**Published:** 2025-02-05

**Authors:** Jixiang Zhao, Changjia He, Haoqun Xie, Yunzhi Zou, Zeming Yan, Jingen Deng, Yizhi Du, Wenzhuo Yang, Xiangheng Zhang

**Affiliations:** State Key Laboratory of Oncology in South China, Guangdong Provincial Clinical Research Center for Cancer, Sun Yat-Sen University Cancer Center, Guangzhou 510060, China; zhaojx@sysucc.org.cn (J.Z.); hecj@sysucc.org.cn (C.H.); jiehq@sysucc.org.cn (H.X.); zouyz@sysucc.org.cn (Y.Z.); yanzm@sysucc.org.cn (Z.Y.); dengje@sysucc.org.cn (J.D.); duyz@sysucc.org.cn (Y.D.); yangwz@sysucc.org.cn (W.Y.)

**Keywords:** glioma, nutrients, risk factor

## Abstract

Background: Gliomas, particularly high-grade gliomas such as glioblastoma, represent a major challenge due to their poor prognosis. While dietary factors have been proposed as potential modulators of glioma risk, causal inference has been hindered by confounding and reverse causality in observational studies. This study employs Mendelian randomization to investigate the causal relationship between dietary factors and glioma risk. Methods: A two-sample MR framework was applied, utilizing genome-wide association study data for 22 dietary exposures and glioma risks, including both GBM and non-GBM subtypes. Instrumental variables (genetic variants) were identified for each dietary factor to address confounding and pleiotropy. Causal inference was conducted using inverse-variance weighted regression, complemented by MR-Egger and MR-PRESSO analyses to assess and correct for potential pleiotropy. Results: A positive causal association was observed between the intake of cooked vegetables and the GBM risk (OR = 6.55, 95% CI: 1.86–23.12, *p* = 0.00350). While alcohol intake demonstrated a protective effect for non-GBM risk (OR = 0.770, 95% CI: 0.61–0.97, *p* = 0.029), beer was substantially linked to an increased risk of non-GBM gliomas (OR = 4.82, 95% CI: 1.84–12.59, *p* = 0.0014). Other dietary factors did not exhibit significant causal associations. Conclusions: These findings suggest that certain dietary factors, including cooked vegetable intake, beer consumption, and alcohol intake, may exert a causal influence on glioma risk. This study provides new insights into the potential dietary determinants of glioma and underscores the need for further investigation into modifiable risk factors for glioma prevention.

## 1. Introduction

Gliomas represent the most prevalent malignant primary brain tumors, accounting for around 80% of malignant primary tumors in adults [1]. It is commonly considered one of the most dreadful tumors in the central nervous system (CNS) with cancerous glial cells that proliferate rapidly. The annual new case incidence rate is approximately 4–5 per 100,000 people [2]. Gliomas constitute a subset of neuroepithelial tumors characterized by histological features resembling those of normal glial cells, including astrocytes, oligodendrocytes, and ependymal cells [3]. According to the 2021 WHO classification of central nervous system tumors, gliomas are graded on a scale of I-IV, with grades I-II considered low-grade gliomas and grades III–IV defined as high-grade gliomas [4]. Among these, glioblastoma (GBM) is the most common and aggressively invasive malignant tumor of the central nervous system, with a dismal 5-year survival rate of only 6.8% [5]. Despite advances in therapeutic approaches, survival rates for high-grade gliomas remain disappointingly poor. Thus, there is an urgent need to explore modifiable risk factors that may help reduce glioma incidence and improve patient prognosis.

Diet is a changeable factor that can affect the risk and progression of cancer by impacting cellular metabolism, inflammation, and other regulatory pathways within the body through its effect [6,7]. Unhealthy diets, such as high-fat and high-sugar foods, increase the production of reactive oxygen species, thereby exacerbating cellular oxidative damage. Antioxidant-rich foods, such as fruits, and whole grains, on the other hand, can reduce levels of oxidative stress by providing antioxidants such as vitamin C and vitamin E [8]. For example, catechins in green tea and anthocyanins in blueberries have been shown to significantly reduce oxidative stress [9]. Because the brain consumes more oxygen than other tissues, it is considered to be vulnerable to oxidative stress (OS) damage, which may promote brain cancer formation [10]. Oxidative stress results from an imbalance between ROS and reactive nitrogen species (RNS). When ROS are in excess, oxidative stress conditions occur in the body. This process is associated with oncogene activation, enhanced metabolism, and mitochondrial dysfunction. In addition, oxidative stress leads to free radical-induced DNA changes that result in genomic instability [11]. Several studies have revealed an association between OS and brain tumor development, and the ROS-mediated activation of the hypoxia-inducible factor (HIF) has been linked to brain tumor development. The activation of HIF-1α and HIF-1α (EPAS1) expression has been observed in glioblastoma stem cell-like cells, and HIF triggers ROS formation via the simultaneous activation of NADPH oxidase. HIF regulates a variety of factors, including VEGF, pyruvate dehydrogenase kinase 1 (PDK1), etc., and is involved in cell proliferation, survival, and angiogenic metabolism [12,13]. In addition, HIF has been shown to regulate apoptosis and cell cycle pathways by modulating multiple transcription factors [14]. The greater the OS-mediated damage, the greater the risk of brain cancer. Secondly, diet can also play a role in tumor progression by altering the inflammatory response microecology. Different dietary patterns can modulate the body’s level of inflammation. Long-term intake of high-fat, high-sugar, and high-calorie diets triggers low-grade chronic inflammation, and such diets activate inflammatory pathways by promoting the release of inflammatory factors (e.g., TNF-alpha, IL-6) and oxidative stress production in adipose tissue [15,16]. In turn, chronic inflammation is thought to be able to promote glioma formation and progression through multiple mechanisms. For example, pro-inflammatory cytokines in the tumor microenvironment (e.g., IL-6, TNF-α, and IL-1β) can activate signaling pathways (e.g., NF-κB and STAT3) that promote tumor cell proliferation, invasive capacity, and resistance to therapy. In addition, immune cells infiltrated in gliomas (e.g., microglia and macrophages) can secrete pro-inflammatory factors, which further intensify the inflammatory response, creating a vicious circle. On the other hand, chronic inflammation also promotes glioma development and progression by inducing DNA damage and suppressing anti-tumor immune responses [17,18,19].

Numerous studies have demonstrated robust associations between various cancer types and dietary variables [8,9,10,11]. For instance, the regular consumption of beverages like tea and mate at scalding hot temperatures has been linked to an elevated risk of esophageal cancer [20,21]. Recent epidemiological research has increasingly investigated the makeup of diet and nutrient intake as potential factors influencing glioma risk [22,23]. A meta-analysis indicated that a greater intake of vegetables may have a protective effect on glioma, while higher fruit consumption may exert a protective influence on glioma risk in Asian populations [22]. The results of research on the influence of diet on glioma risk remain ambiguous due to differences in study design methodologies [22,23,24,25]. The associations observed in observational studies may not represent causal relationships, but could instead arise due to inherent methodological biases within the study designs [26]. Therefore, rigorous and comprehensive research using stringent methodologies is necessary to draw clear causal conclusions about the relationship between diet and gliomas.

Mendelian randomization utilizes genetic variants as natural instrumental variables to investigate causal inference between an exposure and outcome and has been widely used in epidemiological and etiological studies. As genetic variants are randomly allocated at fertilization, Mendelian randomization can help mitigate issues like reverse causation and residual confounding that tend to influence observational research. The random assortment of alleles during gamete formation and conception mimics the ideal setting of a randomized controlled trial, thus strengthening causal claims compared to observational epidemiology [27]. Large genome-wide association studies (GWASs) furnish a wealth of genetic variants that can serve as instrumental variables, enhancing analytical precision compared to investigations relying on only one or a few variants. Consequently, through Mendelian randomization studies, potential causal relationships between modifiable risk factors and diseases can be elucidated. To discern the causal influence of diet on gliomas risk, we employed bidirectional two-sample Mendelian randomization, leveraging summarized data from 22 diet-related and 8 glioma genome-wide association studies. This facilitated the examination of associations between single nucleotide polymorphisms (SNPs) related to 22 common dietary exposures and the risk of gliomas.

## 2. Materials and Methods

### 2.1. Study Design

This study seeks to investigate the potential causal relationship between diet and glioma using a two-sample Mendelian randomization (MR) approach. We conduct a secondary analysis using genome-wide association study (GWAS) summary statistics from the UK Biobank, without accessing individual-level data. In this work, we acquired exposure and outcome data from publicly accessible databases and subsequently identified instrumental variables. Mendelian randomization analyses were conducted utilizing these instrumental factors as mediators. Sensitivity analyses were performed to enhance the robustness of our results. Ethical approval was unnecessary as the analysis solely employed publicly available aggregate data.

### 2.2. Potential Common Dietary Exposures

In this study, we summarize 22 common dietary habits. Fifteen common food intakes are beef, bread, cereal, cheese, cooked vegetables, dried fruit, fresh fruit, lamb/mutton, non-oily fish, oily fish, pork, poultry, processed meat, salad/raw vegetables, and salt added to food. Seven common beverage intakes are water, hot drinks, tea, coffee, alcohol, beer, and red wine. Information regarding the collection, curation, and analysis of these raw GWAS data can be found in the original documentation from the MRC Integrative Epidemiology Unit (MRC-IEU) (https://www.ebi.ac.uk/gwas/downloads/summary-statistics) (accessed on 26 June 2024). Details on accessing the raw data from UK Biobank and the processing steps taken to quality control, impute, and analyze the genetic variants can provide important context for interpreting the results of this study.

### 2.3. GWAS Summary Statistics of Glioma as Outcomes

GWAS data of glioma were retrieved from a meta-analysis of eight independent GWAS datasets comprising over 10 million directly genotyped and imputed SNPs [28]. A total of 12,488 glioma cases and 18,169 controls of European ancestry were included. Cases were dichotomously stratified into glioblastoma (GBM; 6183 cases) and non-glioblastoma (LGG; 5820 cases) histological subtypes.

### 2.4. Instrumental Variables Selection Process

We performed a two-sample Mendelian randomization strategy that integrates genetic association estimates from various variants to evaluate the causal link between diet and glioma. The MR methodology assumes that SNPs utilized as instrumental variables (IVs) are associated solely with the risk factor of interest and are independent of potential confounders or alternative causal pathways. Furthermore, genetic variation is closely related to exposure. Moreover, to precisely estimate the size of the causal effect, associations must be linear and not influenced by statistical interactions or moderation effects [29]. The cutoff thresholds include a minor allele frequency greater than 0.05 and a SNP call filter greater than 0.95. To screen for SNPs to be utilized as instrumental variables, we selected SNPs that were strongly associated (*p* < 5 × 10^−8^) with dietary habits to help ensure precise causal effect estimation, and this threshold has also been applied in numerous studies. We implemented a clumping procedure to prune SNPs according to linkage disequilibrium (LD) within their respective genomic regions. This prevented the instrumental SNPs from being in LD with one another, which could potentially produce misleading findings. Through this clumping process, SNPs were selected that presented LD below a threshold of r^2^ < 0.001 within 10 kb windows. To establish a robust association between the instrumental variables and exposure factors, we evaluated the F-statistic for each SNP. The F-statistic (F = β^2^/se^2^) quantifies the strength of association between each SNP and exposure and eliminates SNPs with F < 10 to mitigate weak instrument bias [30]. All SNPs presented an F-statistic exceeding 10, satisfying the standard threshold for reliable instrumental variables [31]. The F-statistic for each SNP is calculated using the following formula:$$F=\frac{R^{2}}{1-R^{2}}\times \left(N-2\right)$$

The variance of exposure explained by instrumental variables is calculated using the following formula:$$R^{2}=\frac{\beta^{2}}{\left(\beta^{2}+{se}^{2}+N\right)}$$

In the aforementioned formula, *R*^2^ represents the variance of exposure explained by instrumental variables, *N* denotes the sample size, *β* signifies the effect size of genetic variation in dietary habits, and se indicates the standard error of *β*.

### 2.5. Two-Sample MR Analysis

We conducted two-sample Mendelian randomization studies utilizing the TwoSampleMR R package (https://mrcieu.github.io/TwoSampleMR) (accessed on 28 June 2024) in R (version 4.1.2) to investigate the causal relationship and directionality between food and glioma. Forward Mendelian Randomization analysis was conducted using each SNP as the exposure and glioma as the outcome.

### 2.6. Statistical Analysis

In the MR analysis of two samples, we utilized IVW with random effects as the primary method, and Mendelian randomization-Egger (MR-Egger), weighted median (WM), simple mode, and weighted mode as supplementary methods to assess the causal relationship between diet and glioma. The IVW method employed a meta-analysis method, integrating the Wald ratio estimates of causal relationships generated by various SNPs. It provides reliable assessments of the causal effects of exposure on outcomes in the absence of horizontal pleiotropy [32]. WM assumes that a minimum of half of the genetic instruments are valid, deriving the causal effect estimate from the median of the IV weights [33]. The simple model is an approach for evaluating robustness across multiple effects, while the weighted mode is particularly sensitive to rigorous throughput collection [34,35]. MR-Egger regression introduces an intercept term to initially estimate pleiotropy. This estimated pleiotropy parameter is then used to adjust the causal effect estimate. Moreover, MR-PRESSO is employed to identify and account for horizontal pleiotropic effects through the exclusion of outlier instrumental variables [36]. An MR-Egger intercept *p*-value above 0.05 suggests no significant horizontal pleiotropy. When outliers were found, we recalculated the causal estimate with MR-PRESSO and reported this as our primary result. We also calculated heterogeneity in causal estimates across specific variants using Cochran Q statistics to evaluate the robustness of our MR analysis [37]. In our analysis, Cochran Q was not significant at the 0.05 level, supporting the stability of our findings. We performed a “leave-one-out” sensitivity analysis to identify potentially influential SNPs, which involved excluding one SNP at a time and performing an inverse-variance weighted random effects meta-analysis on the remaining SNPs. Moreover, we further assessed potential biases and heterogeneity by employing a leave-one-chromosome-out (LOCO) analysis. This approach enhances the robustness of the results by ensuring that no single chromosome or SNP cluster disproportionately influences the findings. Additionally, heterogeneity was also further assessed through forest plots and scatter plots. We also generated a heatmap and a bubble plot to visually summarize the associations between dietary factors and gliomas.

We presented results as odds ratios (ORs) with 95% confidence intervals (CIs) for each logOR change for each diet. We calculated ORs with 95% CIs per 1 logOR increase in dietary factors. This quantified the relationships between diets with glioma risk. We applied a Bonferroni correction to account for multiple comparisons across dietary factors. Associations with *p* < 0.05 were considered suggestive of a causal relationship. Results meeting a stricter genome-wide significance of *p* < 0.00227 (i.e., 0.05 divided by 22 diets evaluated) provided stronger evidence and a significant association.

## 3. Results

### 3.1. Genetic Instruments for 22 Dietary Habits

Table 1 and Table 2 provides a summary of the 30 GWASs with 22 GWASs of diet habits and 8 GWASs of glioma. The number of SNPs analyzed for associations with each diet habit and glioma risk varied between studies, ranging from 7 to 88 SNPs. The F-statistics of each identified SNP exceeded the threshold of 10, suggesting that all SNPs were robust instrumental variables (Appendix A). To facilitate the clear presentation of results, we categorized dietary habits into two subgroups: food (including cheese, beef, pork, poultry, lamb/mutton, oily fish, non-oily fish, processed meat, cereal, bread, cooked vegetables, fresh fruit, dried fruit, salad/raw vegetables, and salt added to food) intake and beverage (including water, tea, alcohol, beer, red wine, coffee, and hot drinks) consumption. Meanwhile, to explore the possibility that diet may be associated with glioma in a subtype-specific manner, we investigated causal effects in GBM and non-GBM subtypes separately (Appendix A).

### 3.2. Causal Association of Food Intake with Glioma

We leveraged Mendelian randomization to explore the causal associations between each of the 15 food intakes and gliomas (including GBM and non-GBM) by performing the IVW approach under a random-effects model. The summary of causal effects of food habits on different types of gliomas is shown in Figure 1, Figure 2, Figure 3, Figure 4, Figure 5 and Figure 6. The intake of cooked vegetables was positively associated with GBM (OR = 6.55, 95% CI: 1.86–23.12, *p* = 0.00350) (Figure 3 and Figure 7). In cooked vegetable intake with glioma and non-GBM, the results of the IVW method and MR Egger method showed a contrary trend (Appendix A). The results of the IVW method and weighted mode method showed a contrary trend in processed meat intake with non-GBM. There is pleiotropy (*p* < 0.05) between cheese and non-GBM (Appendix A). In addition, no causal relationships were found between other food intakes and glioma or its subtypes (Figure 1, Figure 2 and Figure 3; Appendix A).

### 3.3. Causal Association of Beverage Consumption with Glioma

We also employed Mendelian randomization to explore the potential causal links between the consumption of seven beverage types and glioma (including GBM and non-GBM) using the IVW method under a random-effects model. Figure 1 summarizes the estimated causal effects of various dietary intake patterns on different glioma subtypes. The intake of alcohol was negatively correlated with non-GBM (OR = 0.770, 95% CI: 0.61–0.97, *p* = 0.029) (Figure 1 and Figure 7). However, when performing a leave-one-chromosome-out (LOCO) analysis, we observed that the association between alcohol consumption and non-GBM was significantly influenced by SNPs located on chromosome 16. The LOCO analysis revealed marked differences when excluding chromosome 16, which notably affected the result of the association between alcohol intake and non-GBM risk (Appendix A). Beer consumption showed a significant positive association with non-GBM (OR = 4.82, 95% CI: 1.84–12.59, *p* = 0.0014) (Figure 3 and Figure 7). In beer intake with glioma, the results of the IVW method and MR Egger method showed a contrary trend. In addition, no causal relationships with glioma, GBM, or non-GBM were found for other beverage intakes (Figure 3, Figure 4, Figure 5 and Figure 6; Appendix A).

### 3.4. Reverse MR Analysis Between Glioma Risk and Positive Dietary Factors

We conducted a reverse Mendelian randomization analysis to explore the potential association between genetically predicted glioma risk and positive dietary factors. The analysis revealed no evidence of a causal relationship between glioma risk and positive dietary factors (Appendix A).

## 4. Discussion

This study utilizes a two-sample MR method to characterize the potential causal relationship between dietary factors and the risk of glioma, as well as its subtypes. In the presence of outliers detected by the MR-PRESSO method, we excluded the outliers and re-evaluated the repeated MR analyses, using the new results from the IVW method as the basis for assessing causal relationships. A notable positive correlation was identified between beer drinking and low-grade gliomas. Furthermore, we identified possible correlations between cooked vegetables, lamb, alcohol, and hot beverages and gliomas. These findings suggest that certain dietary factors may influence glioma risk, offering insights into preventive strategies.

The present research indicates a possible negative correlation between alcohol use and non-GBM, implying a preventive effect. Moreover, our LOCO analysis provided additional insight into the relationship between alcohol consumption and non-GBM glioma. We observed that SNPs located on chromosome 16 had a significant impact on the association between alcohol intake and non-GBM risk. The LOCO analysis revealed that when chromosome 16 was excluded, the observed association between alcohol consumption and non-GBM glioma was notably altered. This finding suggests that genetic variation in this specific chromosomal region may be contributing to the relationship between alcohol and non-GBM glioma risk. This highlights the importance of considering specific genetic loci when interpreting Mendelian randomization results, as they may have a disproportionate influence on the findings. The identification of such influential SNPs emphasizes the value of conducting more detailed genetic investigations to explore how individual genetic variants, particularly in specific chromosomes, could modulate the effect of dietary factors like alcohol on glioma risk. Further research into these genetic regions may provide valuable insights into the molecular mechanisms underlying the association between alcohol consumption and glioma.

Although alcohol consumption has been widely recognized as a significant risk factor for various cancers, typically regarded as a carcinogen, it may play a protective role in glioma [38,39]. In 2021, a prospective cohort study indicated a significant inverse association between alcohol intake and glioma risk [39]. Another study found that light-to-moderate drinkers had a relative risk of 0.87 compared to non-drinkers, further supporting the potential protective effect of alcohol against glioma [40,41]. Alcohol may influence glioma risk via its impact on the neurological system and can regulate immune cell function in the central nervous system, therefore affecting neuroinflammation [35,36,42,43]. In addition, alcohol may regulate the expression of immunosuppressive cytokines, such as IL-6 and TNF-α, which could affect the glioma microenvironment and inhibit tumor initiation and progression [42,44]. Nonetheless, this protective effect was not seen for glioblastoma (GBM). The highly immunosuppressive microenvironment in GBM, characterized by regulatory T-cell enrichment, PD-L1 overexpression, and increased inhibitory cytokines such as TGF-β, may neutralize any immunomodulatory benefits conferred by alcohol intake [45].

Moreover, some studies suggest that moderate alcohol consumption may promote the activity of antioxidant enzymes, such as superoxide dismutase and glutathione peroxidase, thereby enhancing the body’s antioxidant capacity and reducing intracellular oxidative damage [46]. Glioma cells are typically in a state of high oxidative stress, which can lead to DNA damage, apoptosis, and alterations in the tumor microenvironment [47]. Therefore, we hypothesize that the inhibitory effect of alcohol consumption on oxidative stress mechanisms may support our findings.

Meanwhile, it has been found that beer consumption is significantly positively correlated with the risk of non-GBM and shows a suggestive positive correlation with glioma risk, but it is unrelated to GBM. As one of the most widely consumed beverages globally, it is unfortunate that commercially available beer has been shown to contain physical, chemical, and/or microbial contaminants, all of which pose potential health risks to humans and animals. The pathogenic effects of beer may be associated with its specific ingredients and brewing processes. Major chemical contaminants identified during beer production include acrylamide, biogenic amines, heavy metals, mycotoxins, nitrosamines, heterocyclic amines (HAs), and other derivatives [48]. For instance, acrylamide, one of the neurotoxic components found in beer, induces the formation of its primary DNA adduct, N7-GA-Gua, which can generate apurinic/apyrimidinic (AP) sites, leading to G→T transversions. Additionally, another secondary adduct, N1-GA-Ade, exhibits significant miscoding potential, making acrylamide highly mutagenic. These molecular properties of acrylamide are closely associated with tumor initiation and progression [49]. Furthermore, beer contains malt-derived compounds, pigments, and heterocyclic amines, which can activate the epidermal growth factor receptor (EGFR)-MAPK and PI3K-Akt signaling pathways, thereby promoting tumor cell proliferation and survival [50]. Upon activation, EGFR initiates downstream signaling pathways, including the MAPK and PI3K-Akt pathways, which promote cell proliferation, migration, and survival. The MAPK pathway resists apoptosis by regulating cell survival-related proteins such as Bcl-2, thereby enhancing tumor cell survival. The PI3K-Akt signaling pathway plays a crucial role in cell survival, proliferation, and metabolism, suppressing apoptosis-related pathways and promoting cell survival, which helps tumor cells adapt to adverse environments [51,52,53]. Additionally, the high sugar content and certain additives in beer may alter the composition of the gut microbiota, indirectly influencing systemic inflammation and immune status [54]. The gut microbiota, as a producer of short-chain fatty acids (SCFAs) such as acetate and butyrate, can stimulate the development of regulatory T cells. These cells contribute to the immunosuppressive environment of glioblastoma (GBM) by producing interleukin-10 (IL-10) and transforming growth factor-β (TGF-β), potentially making the gut microbiota a risk factor for gliomas [55]. In addition, the high sugar content and certain additives may alter gut microbiota composition, indirectly affecting systemic inflammation and immune status [55]. Unlike red wine, which contains antioxidant components such as polyphenols and resveratrol, beer generally lacks these protective agents, potentially reducing its capacity to mitigate oxidative stress and inflammation. These chemical and biological characteristics may contribute to its observed positive association with non-GBM risk. When discussing the differential pathogenic effects of beer on GBM and non-GBM tumors, it is necessary to consider their distinct biological characteristics. Non-GBM tumors, due to their slower growth rate and longer cell division cycle, may be more susceptible to the effects of beer and its metabolites, such as acetaldehyde. Additionally, non-GBM tumors may exhibit differences in the activity of alcohol-metabolizing enzymes during alcohol metabolism, leading to varied responses to alcohol and its metabolites [56]. Studies have shown that certain types of non-GBM tumors may exhibit higher sensitivity to carcinogens like acetaldehyde. In contrast, GBM, as a highly aggressive tumor, may demonstrate greater adaptability, allowing its cells to tolerate the metabolic stress associated with beer consumption [57]. Furthermore, the microenvironment of GBM is typically more complex and may employ various mechanisms, such as immune evasion and angiogenesis, to counteract the potential negative effects of beer. These factors might explain why our study shows a weaker correlation between beer consumption and the impact on GBM [39,58,59,60]. Of course, studies regarding the influence of beer on gliomas are scarce, highlighting the necessity for additional research. The study conducted on the correlation between beer consumption and tumors is insufficient, warranting additional examination.

Our study also indicates suggestive associations between the consumption of cooked vegetables and GBM. Multiple mechanisms may be involved in this process. Vegetables contain abundant nitrate, which can transform into nitrosamines under conditions of heat processing [61]. Nitrosamines are potent carcinogens and mutagens in humans, inducing carcinogenesis through mechanisms such as DNA alkylation that leads to mutations and the generation of oxidative stress damaging cellular components [62,63,64]. It can form DNA adducts by adding alkyl groups to DNA bases, resulting in mispairing during replication and the potential activation of oncogenes or the potential inactivation of tumor-suppressor genes [65,66]. Studies also suggest that polycyclic aromatic hydrocarbons (PAHs) and heterocyclic amines (HCAs), generated under high-temperature cooking conditions, may further contribute to DNA damage and promote tumorigenesis [67,68]. Further analysis suggests that high-temperature cooking methods, such as prolonged boiling or frying, substantially increase the formation of nitrosamines and acrylamide [69,70]. The use of cooking oils in high-temperature frying may additionally produce lipid oxidation products (LOPs), which may contribute to carcinogenesis [71]. Moreover, the oxidative stress induced by nitrosamines can lead to lipid peroxidation and protein oxidation, contributing to cellular damage and cancer development [72,73]. It has been proven to exhibit carcinogenic and mutagenic properties in humans, leading to a range of diseases including gastric, colorectal, and esophageal cancers [74]. Additionally, acrolein can be generated during the thermal processing of foods, including vegetables, through the Maillard reaction process [75]. The International Agency for Research on Cancer (IARC, 2020) has classified acrolein as a possible human carcinogen (Group 2A), based on “sufficient” evidence of carcinogenicity in experimental animals and “strong” mechanistic evidence [76]. This classification suggests that, although direct evidence from epidemiological studies in humans is limited, findings from experimental animals and potential carcinogenic mechanisms support its carcinogenic potential [56,76]. It has been reported that acrolein may promote the development of high-fat diet-induced colon tumors in mice by activating the RAS/MAPK pathway [77]. When this pathway is activated by acrolein, it may lead to uncontrolled cell proliferation, thereby accelerating tumor development [77,78]. Acrolein may also accumulate in the brain and promote the formation and spread of GBM by altering the neuroinflammatory environment [79]. Studies have shown that vegetables can produce hydrogen peroxide (H_2_O_2_) during the cooking process [80], suggesting that cooked vegetables may influence the tumor microenvironment of GBM through oxidative stress pathways. Various processes, including oncogene activation, enhanced metabolism, and mitochondrial dysfunction, are associated with elevated H_2_O_2_ levels. Furthermore, oxidative stress mediated by H_2_O_2_ can lead to free radical-induced DNA alterations, resulting in genomic instability [81], thereby promoting the initiation and progression of GBM. Additionally, we hypothesize that the differential effects of cooked vegetables on GBM and non-GBM tumor risk may also be related to their biological differences. GBM cells, due to their increased sensitivity to oxidative stress, may be more affected by changes in nutrient composition [82,83]. In contrast, non-GBM tumors, such as meningiomas, exhibit relatively lower sensitivity to oxidative stress and neuroinflammatory environments [84]. This difference in biological characteristics may provide insights into the variable impact of oxidative stress induced by dietary factors on GBM and non-GBM tumor types.

Furthermore, the potential health effects of vegetable consumption may be influenced by various factors, including the type of vegetable, cooking methods, and an individual’s digestive and absorption capacity. The impact of consuming cooked vegetables may not depend on a single factor but could also be closely related to an individual’s genetic background, dietary habits, and lifestyle.

The findings of this study hold practical significance for clinicians to enhance health education for glioma patients. Avoiding beer intake can reduce the risk of glioma in high-risk populations. Although current evidence suggests an inverse association between alcohol consumption and the risk of low-grade glioma, it is crucial to highlight that the impact of alcohol intake may vary among individuals. Additionally, the carcinogenic potential of alcohol is influenced by factors such as the level of consumption, individual metabolic profiles, and genetic predispositions. Additional investigation is required to elucidate its possible molecular mechanisms and validate its clinical importance. Future research endeavors will aid in glioma prevention and have substantial ramifications for public health.

In this study, we also observed that the IVW and MR-Egger methods produced divergent trends in the associations between the consumption of cooked vegetables and beer with glioma risk. Residual pleiotropy may be the primary cause of these discrepancies. MR-Egger accounts for pleiotropy, providing more robust estimates, whereas the IVW method assumes that instrumental variables affect the outcome solely through the exposure and may consequently overlook residual pleiotropic effects. When pleiotropy is not adequately addressed, MR-Egger may yield more accurate estimates. Additionally, limitations inherent to these statistical methods could contribute to the inconsistency. IVW is most accurate when instruments are free of pleiotropy, whereas MR-Egger can adjust for pleiotropy but may be less precise with weak instruments or high data heterogeneity. Therefore, the divergence between IVW and MR-Egger underscores the importance of employing multiple methods to ensure consistency. By cross-validating various MR approaches, we gain a more comprehensive understanding of the causal relationship between dietary factors and glioma risk, thereby enhancing the reliability of our findings.

The strengths of this study lie in exploring the relationship between various dietary intakes and gliomas through MR. The MR design itself is less susceptible to residual confounding. We addressed potential pleiotropy by employing multiple MR methods, GWAS databases, and excluding SNPs related to various dietary factors. Therefore, our results are less likely to be affected by horizontal pleiotropy. Another strength of this study is that genetic variation in dietary intake and gliomas was derived from summary-level data of large-scale GWAS, encompassing over 30,000 participants. The robustness of our findings was further supported by investigating horizontal pleiotropy using the MR-PRESSO method.

This research possesses specific limitations. Initially, both subjects in the exposure and outcome data are of European descent; due to inherent genetic variations among different ethnic groups and differences in dietary habits, our findings cannot be fully generalized to other racial populations. So, additional research is required to ascertain whether our findings may be extrapolated to other ethnicities. Furthermore, we are unable to ascertain a dose–response association between dietary variables and gliomas. Additionally, while this study analyzed 22 individual dietary factors, it may not fully capture the broader impact of overall dietary patterns, such as the Mediterranean or Western diets. Overall, dietary patterns reflect complex interactions among various foods and nutrients, which could have distinct effects on glioma risk. Future studies should aim to incorporate dietary pattern scores (e.g., Mediterranean diet score or Western diet) as exposures in Mendelian randomization analyses to better understand the relationship between comprehensive dietary patterns and gliomas. Furthermore, the absence of sex-specific summary-level GWAS data precluded us from performing stratified analyses by gender. Future GWAS investigations concerning dietary components should distinguish between male and female participants. Ultimately, even with the application of control measures, IVs may remain vulnerable to unmeasured confounding that could influence the outcomes. Although the F-statistics exceed 10, indicating that the instrumental variables are sufficiently strong, this outcome does not necessarily imply a reduction in bias but rather an increased probability of a smaller bias. Moreover, with smaller effect sizes, the presence of weak instrument bias remains a possibility. Additionally, residual confounding could lead to biased interpretation of results. Future studies should employ more sophisticated multivariable Mendelian randomization (MR) methods for further validation.

Future research should further explore specific molecular pathways involved in the observed associations between dietary factors and gliomas. Additionally, there is a need for an in-depth investigation into the mechanisms by which alcohol and its metabolites contribute to glioma occurrence, as well as the relationship between drinking habits and gliomas. Such studies can guide the formulation of public health policies and the implementation of individual preventive measures. These future research efforts will support glioma prevention and provide more specific recommendations for public health policy to prevent the occurrence of gliomas and related diseases. In general, these research findings are crucial for clinical doctors to enhance health education for glioma patients. They furnish new insights into the intricate link between nutrition, lifestyle, and glioma risk, and present useful guidance for next epidemiological investigations and therapeutic trials. Apart from dietary factors, other potential carcinogenic factors such as environmental exposure and genetic factors should also be considered to comprehensively assess the risk factors associated with glioma occurrence.

## 5. Conclusions

This study provides evidence of potential causal relationships between specific dietary factors and glioma risk, offering new insights into the role of diet in glioma pathogenesis. Notably, our findings reveal a positive association between cooked vegetable intake and GBM risk, which may be attributed to carcinogenic compounds such as nitrosamines and lipid oxidation products generated during high-temperature cooking. Additionally, alcohol consumption demonstrated a protective effect against non-GBM, likely mediated through its immunomodulatory properties and effects on the glioma microenvironment. Conversely, beer consumption was associated with an increased risk of non-GBM, potentially due to its specific brewing byproducts, such as malt-derived compounds, sugar content, and gut microbiota alterations, which may promote neuroinflammation and tumorigenesis.

These findings underscore the complex interplay between diet and glioma risk and highlight the potential for dietary interventions as a strategy for glioma prevention. While our results provide valuable directions for understanding glioma etiology, they also emphasize the importance of considering broader dietary patterns and interactions among dietary components in future studies.

Furthermore, the implications of this study extend to clinical and public health practices. Educating high-risk populations about the potential risks associated with specific dietary factors, such as excessive beer consumption, and the potential protective effects of moderate alcohol intake may contribute to targeted prevention strategies. Future research should aim to validate these findings in diverse populations, explore dose–response relationships, and investigate the molecular mechanisms underlying these associations. Ultimately, this study lays a foundation for the development of evidence-based dietary guidelines that could reduce glioma risk and improve public health outcomes.

## Figures and Tables

**Figure 1 nutrients-17-00582-f001:**
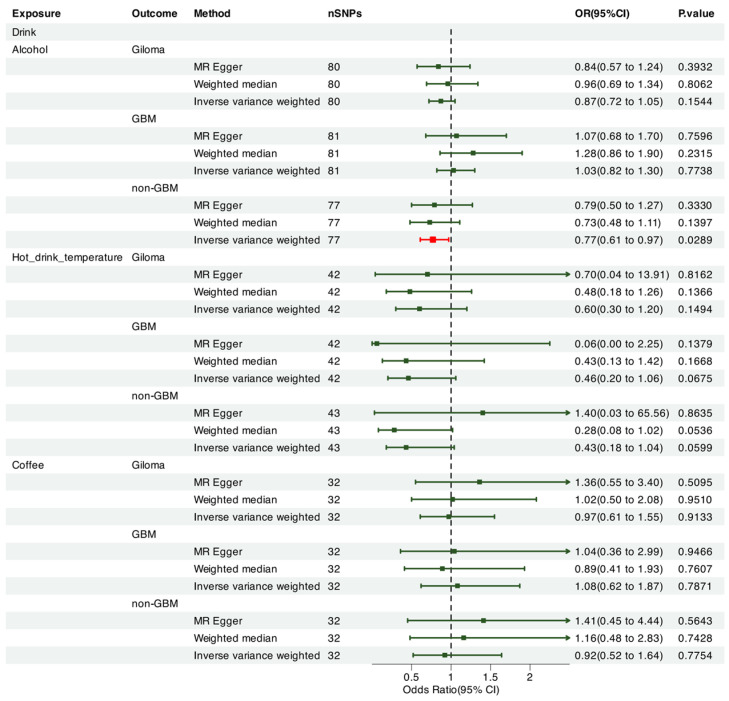
Forrest plot of multi-variable MR estimates for the causal associations between glioma and alcohol, hot drink temperature, and coffee.

**Figure 2 nutrients-17-00582-f002:**
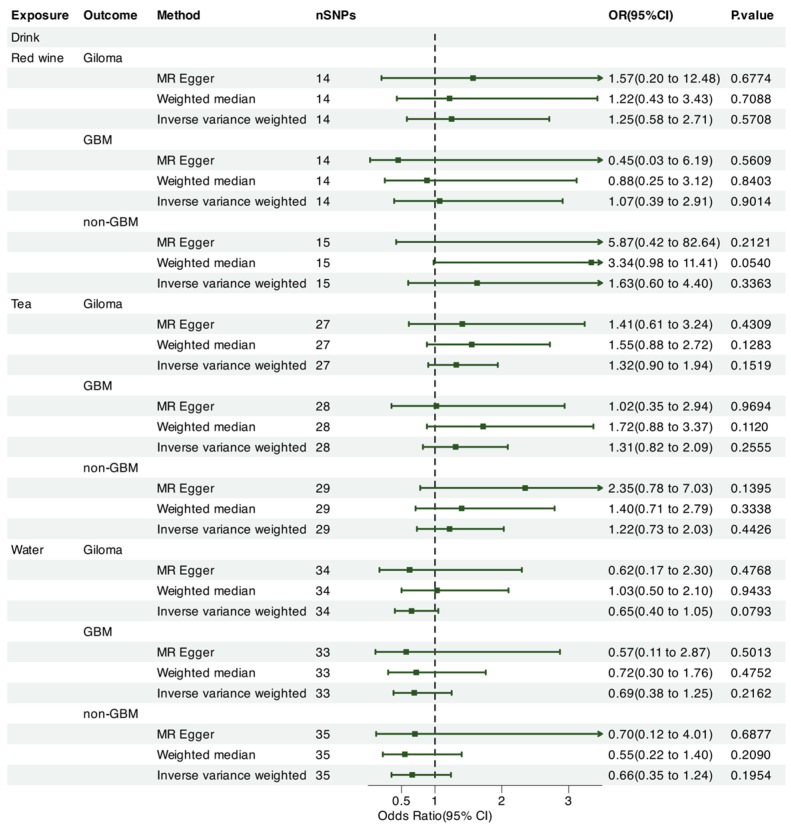
Forrest plot of multi-variable MR estimates for the causal associations between glioma and red wine, tea, and water.

**Figure 3 nutrients-17-00582-f003:**
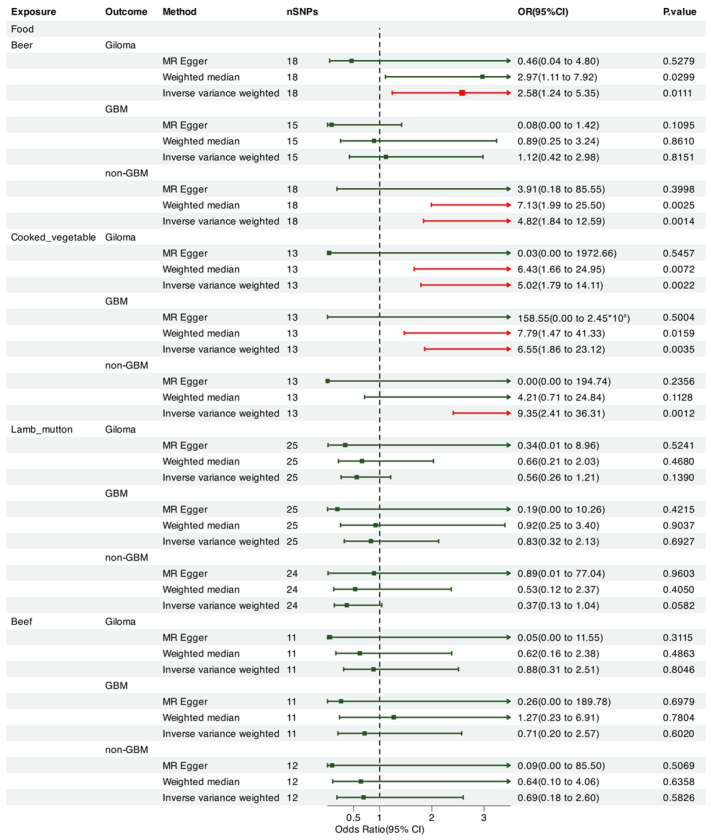
Forrest plot of multi-variable MR estimates for the causal associations between glioma and beer, cooked vegetables, lamb/mutton, and beef.

**Figure 4 nutrients-17-00582-f004:**
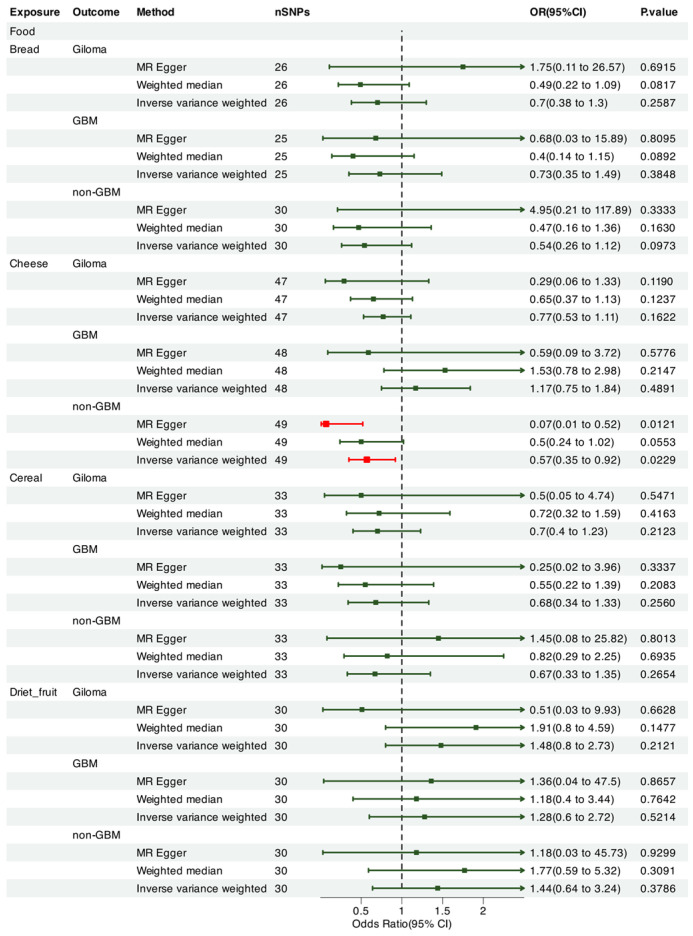
Forrest plot of multi-variable MR estimates for the causal associations between glioma and bread, cheese, cereal, and dried fruit.

**Figure 5 nutrients-17-00582-f005:**
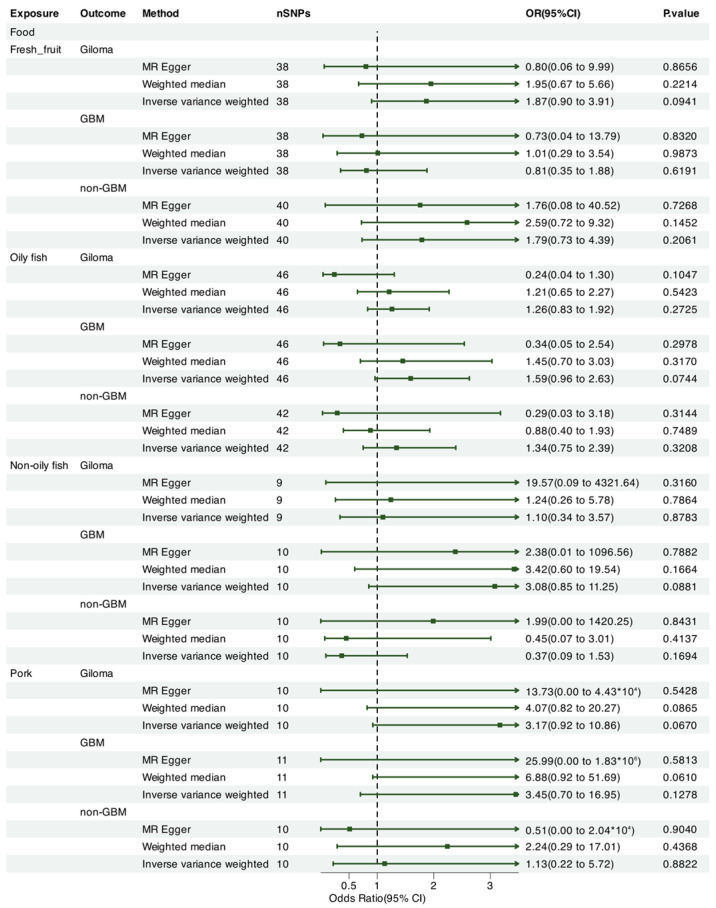
Forrest plot of multi-variable MR estimates for the causal associations between glioma and fresh fruit, oily fish, non-oily fish, and pork.

**Figure 6 nutrients-17-00582-f006:**
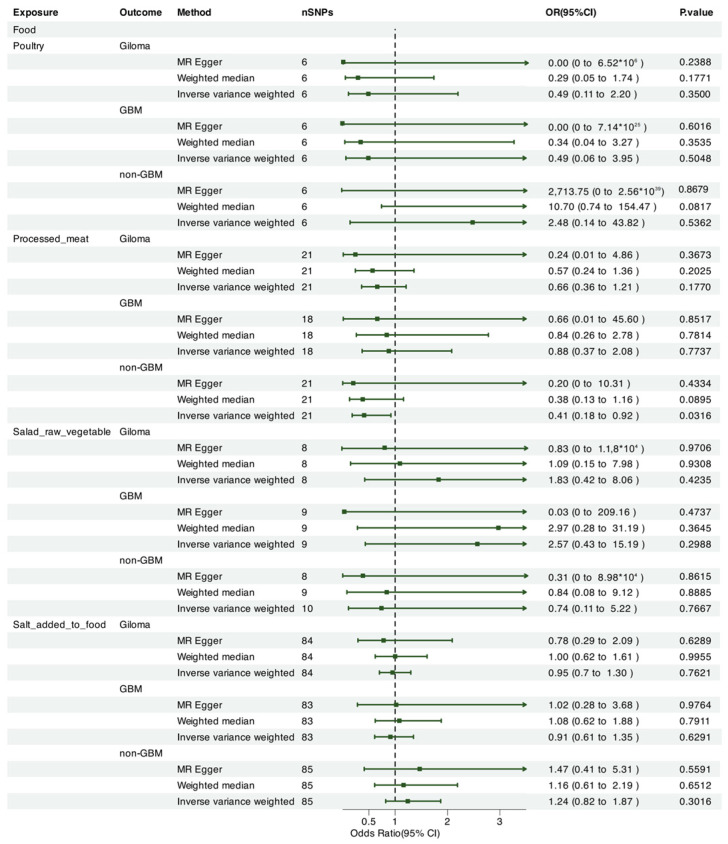
Forrest plot of multi-variable MR estimates for the causal associations between glioma and poultry, processed meat, salad/raw vegetables, and salt added to food.

**Figure 7 nutrients-17-00582-f007:**
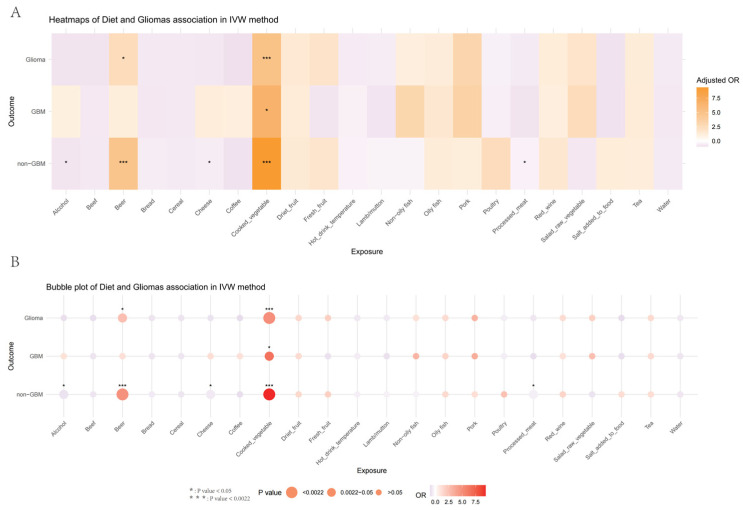
Heatmap and bubble plot of dietary factors and glioma association in IVW method. (**A**) Heatmap of dietary factors and glioma association in IVW method. (**B**) Bubble plot of dietary factors and glioma association in IVW method.(*: *p* value < 0.05; ***: *p* value < 0.0022).

**Table 1 nutrients-17-00582-t001:** Summary datasets on diet.

Types	Trait	ID or PMID	Sample Size	SNPs (N)	Population	Year
Drink	Alcohol	ukb-b-5779	462,346	9851867	European	2018
Drink	Beer	ukb-b-5174	327,634	9851867	European	2018
Drink	Red wine	ukb-b-5239	327,026	9851867	European	2018
Food	Beef	ukb-b-2862	461,053	9851867	European	2018
Food	Bread	ukb-b-11348	452,236	9851867	European	2018
Food	Cereal	ukb-b-15926	441,640	9851867	European	2018
Food	Cheese	ukb-b-1489	451,486	9851867	European	2018
Drink	Coffee	ukb-b-5237	428,860	9851867	European	2018
Food	Cooked vegetables	ukb-b-8089	448,651	9851867	European	2018
Food	Dried fruit	ukb-b-16576	421,764	9851867	European	2018
Food	Fresh fruit	ukb-b-3881	446,462	9851867	European	2018
Drink	Hot drink temperature	ukb-b-14203	457,873	9851867	European	2018
Food	Lamb/mutton	ukb-b-14179	460,006	9851867	European	2018
Food	Non-oily fish	ukb-b-17627	460,880	9851867	European	2018
Food	Oily fish	ukb-b-2209	460,443	9851867	European	2018
Food	Pork	ukb-b-5640	460,162	9851867	European	2018
Food	Poultry	ukb-b-8006	461,900	9851867	European	2018
Food	Processed meat	ukb-b-6324	461,981	9851867	European	2018
Food	Salad/raw vegetables	ukb-b-1996	435,435	9851867	European	2018
Food	Salt added to food	ukb-b-8121	462,630	9851867	European	2018
Drink	Tea	ukb-b-6066	447,485	9851867	European	2018
Drink	Water	ukb-b-14898	427,588	9851867	European	2018

**Table 2 nutrients-17-00582-t002:** Summary database on glioma subtypes (all-glioma, GBM, and non-GBM).

PubMID	Database	Trait			
		All-Glioma Cases	GBM Cases	Non-GBM Cases	Normal Cases
17636416	UK-GWAS	631	270	361	2699
21531791	French-GWAS	1423	430	993	1190
26424050	German-GWAS	846	431	415	1310
19578367	MDA-GWAS	1175	652	523	2236
19578367	UCSF-SFAGS	677	511	166	3940
22886559	GliomaScan	1653	903	472	2725
26656478	GICC	4564	2460	1898	3265
19578366	UCSF/Mayo	1519	526	992	804

## Data Availability

All custom code and data supporting this study will be made available upon request to the corresponding author.

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
