# Peer review of "Latent Association Between Diets and Glioma Risk: A Mendelian Randomization Analysis"

_nutrients, 2025, doi:10.3390/nu17030582_

Round 1
Reviewer 1 Report
Comments and Suggestions for Authors
In the manuscript entitled “Latent Association between Diets and Glioma Risk: A Mendelian Randomization Analysis” the authors use Mendelian randomization to investigate the causal relationship between dietary factors and glioma risk, analyzing GWAS data for 22 dietary exposures and glioma risk (GBM and non-GBM). The results show a positive association between consumption of cooked vegetables and GBM risk, while alcohol intake shows a protective effect for non-GBM risk but increases the risk of other types of glioma. Other dietary factors did not show significant associations. The study offers new insights into possible dietary determinants of glioma, indicating the need for further research on modifiable risk factors.
The manuscript is well written, but I add my suggestions below to improve it.
Minor points.
1. In some cases, IVW and MR-Egger methods show conflicting trends (e.g., in the association between cooked and non-GBM vegetables). It would be useful to discuss further the possible reasons for these discrepancies, such as the presence of residual pleiotropy or limitations in the statistical methods.
2. I suggest that the authors include additional sensitivity analyses, such as Leave-One-Chromosome-Out (LOCO) testing, to verify that the results are not influenced by specific SNP variants or systematic biases.
3. The set of 22 dietary factors, although broad, may not fully capture dietary patterns. Authors may consider integrating data on overall dietary patterns (e.g., Mediterranean or Western diet) to investigate broader associations.
4. The authors should expand the discussion on the biological impact of the observed associations. For example, better explain why consumption of cooked vegetables or beer may affect the risk of GBM or non-GBM, exploring potential molecular mechanisms.
Author Response
- 1. In some cases, IVW and MR-Egger methods show conflicting trends (e.g., in the association between cooked and non-GBM vegetables). It would be useful to discuss further the possible reasons for these discrepancies, such as the presence of residual pleiotropy or limitations in the statistical methods.(Response:Thank you for your insightful comment regarding the discrepancies observed between IVW and MR-Egger methods. We have carefully considered this issue and have supplemented our discussion accordingly. We appreciate your valuable suggestion and believe that this addition strengthens the discussion of our findings.; Marked: Paragraph 9, Discussion section. Line 430 to 443. )
- 2. I suggest that the authors include additional sensitivity analyses, such as Leave-One-Chromosome-Out (LOCO) testing, to verify that the results are not influenced by specific SNP variants or systematic biases.(Response:Thank you for your valuable suggestion regarding additional sensitivity analyses. As per your recommendation, we have incorporated the Leave-One-Chromosome-Out (LOCO) method to further assess the robustness of our findings. We have detailed the methodology of the LOCO analysis in the Methods section and integrated the findings into the Rrsults section and Discussion section, where we address any notable discrepancies. ; Marked: 2.6 Statistical analysis, Materials and Methods section; Line 196 to 199. 3.3. Causal association of beverage consumption with glioma, Results section; Line 242 to 246. Paragraph 2, Discussion section; Line 283 to 297. Supplementary Figure 23-30)
- 3. The set of 22 dietary factors, although broad, may not fully capture dietary patterns. Authors may consider integrating data on overall dietary patterns (e.g., Mediterranean or Western diet) to investigate broader associations.(Response:Thank you for your insightful comment. While we agree that analyzing overall dietary patterns, such as the Mediterranean or Western diets, could provide a more comprehensive understanding of dietary influences on glioma risk, we encountered a limitation in terms of the availability of relevant GWAS data for these broader dietary patterns. We have acknowledged this limitation in the Discussion section, where we highlighted the importance of broader dietary patterns in capturing the complex interactions among various foods and nutrients. Overall dietary patterns may have distinct effects on glioma risk that are not fully captured by examining individual dietary factors. Therefore, we suggest that future studies consider incorporating dietary pattern scores (e.g., Mediterranean Diet Score or Western Diet) as exposures in Mendelian randomization analyses. This approach could help better understand the relationship between comprehensive dietary patterns and gliomas and provide more robust insights into the dietary determinants of glioma risk. We appreciate your recommendation, and we believe that incorporating broader dietary patterns in future studies would certainly enrich the understanding of dietary influences on glioma.; Marked: Paragraph 11, Discussion section; Line 458 to 465. )
- 4. The authors should expand the discussion on the biological impact of the observed associations. For example, better explain why consumption of cooked vegetables or beer may affect the risk of GBM or non-GBM, exploring potential molecular mechanisms.(Response:Thank you for your valuable suggestion. We have expanded the Discussion section to further explore the potential biological mechanisms underlying the observed associations, particularly the consumption of cooked vegetables and beer with the risk of gliomas, including GBM and non-GBM.We have integrated these biological insights into the revised Discussion section and believe they provide a more comprehensive understanding of the observed associations between dietary factors and glioma risk.; Marked: Paragraph 4, Discussion section; Line 313 to 319. Paragraph 5, Discussion section; Line 332 to 349. Paragraph 5, Discussion section; Line 355 to 368. Paragraph 6, Discussion section; Line 401 to 414.)
Reviewer 2 Report
Comments and Suggestions for Authors
The criteria for choosing single nucleotide polymorphisms (SNPs), including the F-statistic criterion (>10), are stated but lack adequate detail concerning genomic database coverage and the management of probable SNP overlaps. Incorporating this information would enhance the robustness and clarity of the process.
The introduction doesn't sufficiently address the molecular factors connecting nutrition to glioma risk. Enhancing this section to encompass plausible mechanisms, including oxidative stress, inflammation, and nutrient-specific interactions, would fortify the study's logic.
Figure 1 is inadequately sized and has sufficient resolution, hindering readers' ability to analyze the data properly. The authors should enlarge the figure and improve its quality to guarantee clarity.
Additional figures need to be incorporated to attract readers and deliver a more thorough representation of the results. For instance, scatter plots, heatmaps, or graphical representations of essential findings could enhance the visual attractiveness of the study.
The authors need to incorporate a distinct section addressing the possible constraints of generalizing their findings to other ethnic groups. Emphasizing the study's concentration on European populations and the necessity for validation in varied cohorts will enhance the study's generalizability.
A discussion on residual confounding and weak instrument bias is essential, especially in instances where effect sizes are low. Resolving these challenges would demonstrate the validity of the results and aid in contextualizing the findings.
The conclusion presently understates the study's implications.
Author Response
- 1. The criteria for choosing single nucleotide polymorphisms (SNPs), including the F-statistic criterion (>10), are stated but lack adequate detail concerning genomic database coverage and the management of probable SNP overlaps. Incorporating this information would enhance the robustness and clarity of the process.(Response:We sincerely appreciate your insightful comments. After reviewing recent Mendelian randomization studies published over the past five years, we recognize the issue of sample overlap, particularly from the UK Biobank data, which could potentially lead to an overestimation of causal effects.However, these studies do not address the potential problem of overlapping SNPs, and sample overlap is inherently difficult to completely avoid. Currently, genetic correlation analyses can mitigate this effect, and we believe that sensitivity analyses can also reduce the overestimation caused by sample overlap. To this end, we have incorporated the Leave-one-chromosome-out (LOCO) method into our existing sensitivity analyses. This involves excluding all SNPs on the chromosome containing the SNPs under investigation and then conducting the tests. While the new method has slightly altered our results, the causal inference between dietary habits and glioma remains unchanged. The criteria for SNP selection are as follows: Firstly, we screen for SNPs strongly associated with dietary habits, setting the significance threshold at less than 5*10-8. Secondly, we apply a clumping procedure to eliminate linkage disequilibrium between SNPs, using a threshold of r2 < 0.001 within 10 kb windows. Thirdly, All SNPs must be harmonised with the allele associated with increased exposure to ensure that our risk increase for exposure has an impact on the outcome. Lastly, we use an F-statistic to assess the association between the instrumental variables and exposure, removing any weak instrumental variables with F < 10 to avoid potential biases. We have also supplemented the content in section 2.4. Instrumental Variables Selection Process with additional details on the process of selecting instrumental variables.; Marked: 2.4. Instrumental Variables Selection Process, Materials and Methods section; Line 146 to 148; Line 150; Line 158; Line 160 to 167.)
- 2. The introduction doesn't sufficiently address the molecular factors connecting nutrition to glioma risk. Enhancing this section to encompass plausible mechanisms, including oxidative stress, inflammation, and nutrient-specific interactions, would fortify the study's logic.(Response:We appreciate the reviewer’s insightful comment regarding the need to further address the molecular factors connecting nutrition to glioma risk. In response to this valuable feedback, we have enhanced the introduction of the manuscript to incorporate a more detailed discussion of the plausible molecular mechanisms underlying the observed associations between diet and glioma risk.; Marked: Paragraph 2, Introduction section. Line 47 to 81.)
- 3. Figure 1 is inadequately sized and has sufficient resolution, hindering readers' ability to analyze the data properly. The authors should enlarge the figure and improve its quality to guarantee clarity.(Response:We appreciate the reviewer’s feedback regarding the sizing and resolution of Figure 1. We understand the importance of ensuring that the figures are clear and of sufficient quality for proper analysis. In response, we have enlarged Figure 1 and enhanced its resolution to ensure that the data is presented more clearly. The revised figure now provides a more accessible and visually interpretable representation of the results.; Marked: in Figure 1.)
- 4. Additional figures need to be incorporated to attract readers and deliver a more thorough representation of the results. For instance, scatter plots, heatmaps, or graphical representations of essential findings could enhance the visual attractiveness of the study.(Response:We greatly appreciate the reviewer’s suggestion to incorporate additional figures to enhance the visual appeal and comprehensiveness of our results. In response to this feedback, we have added two new figures: a heatmap and a bubble plot. The heatmap and bubble plot visually represents the associations between dietary factors and glioma risk, making it easier for readers to grasp the strength and direction of these relationships. These additions aim to improve the clarity of our findings and make the study more visually engaging for readers. We believe these new visualizations will better illustrate the key results and provide a more thorough representation of our data. We hope that these updated figures meet your expectations and enhance the overall clarity and impact of the study.; Marked: 2.6 Statistical analysis, Materials and Methods section; Line 200 to 202.)
- 5. The authors need to incorporate a distinct section addressing the possible constraints of generalizing their findings to other ethnic groups. Emphasizing the study's concentration on European populations and the necessity for validation in varied cohorts will enhance the study's generalizability.(Response:We sincerely appreciate the reviewer’s valuable feedback. In response to the suggestion, we have expanded the discussion section to address the potential limitations related to the generalizability of our findings. One limitation of every Mendelian randomization study is the difficulty in generalizing causal relationships to other racial groups. Additionally, dietary habits vary significantly among different races, which highlights the primary shortcoming of our study: the homogeneity of our research population. We eagerly desire to collaborate with researchers from various countries and regions to investigate populations with diverse racial backgrounds. However, due to current resource constraints, we are unable to carry out such studies. Yet, one of our future research directions is to replicate our study in other racial groups to explore whether similar relationships between dietary habits and glioma exist in those populations.; Marked: Paragraph 11, Discussion section. Line 454 to 456.)
- 6. A discussion on residual confounding and weak instrument bias is essential, especially in instances where effect sizes are low. Resolving these challenges would demonstrate the validity of the results and aid in contextualizing the findings.(Response:We appreciate your gracious remarks. It is well recognized that eliminating bias from weak instrumental variables is a challenging endeavor. In our study, we have adopted the common practice of considering instrumental variables with F-statistics less than 10 as weak, a threshold that effectively rules out the majority of such variables. To further ensure the robustness of our causal inferences, we have employed a variety of sensitivity analyses. ; Marked: Paragraph 11, Discussion section. Line 470 to 475.)
- 7. The conclusion presently understates the study's implications.(Response:Thank you for your insightful comment. In response, we have strengthened the conclusion section to more adequately reflect the broader implications of our findings. We have revised the conclusion to highlight the potential causal relationships between specific dietary factors and glioma risk, emphasizing the novel insights into how diet may influence glioma pathogenesis. We have expanded on the findings, such as the positive association between cooked vegetable intake and GBM risk, which may be attributed to carcinogenic compounds formed during high-temperature cooking. We also discuss the protective effect of alcohol consumption against non-GBM and the increased risk associated with beer consumption, linking these to potential mechanisms like immunomodulation, brewing byproducts, and gut microbiota alterations. Additionally, we now emphasize the broader implications of these findings for public health and clinical practices, including the need for educating high-risk populations about dietary risks and benefits. Finally, we have stressed the importance of validating these findings in diverse populations, exploring dose-response relationships, and further investigating the molecular mechanisms involved. We believe that these revisions provide a more comprehensive overview of the study's potential impact, contributing to the development of evidence-based dietary guidelines aimed at glioma prevention.; Marked: Conclusion section. Line 491 to 515.)
Round 2
Reviewer 2 Report
Comments and Suggestions for Authors
Figures 1A to 1F should be displayed separately to enhance visibility and provide a larger view.
Author Response
Comments 1: Figures 1A to 1F should be displayed separately to enhance visibility and provide a larger view.(Response:Thank you for your valuable feedback. In response to your suggestion, we have split Figures 1A to 1F into separate panels (Figures 1 to 6) to enhance their visibility and provide a clearer, larger view. We believe this modification improves the overall presentation of the data and ensures that each figure is easier to interpret.; Marked: 5. Figures and Tables, Results section. Line 264 to 282. Figure 1-6 )